# Few-Shot Continuous Authentication for Mobile-Based Biometrics

Kensuke Wagata  and Andrew Beng Jin Teoh *

School of Electrical and Electronic Engineering, College of Engineering, Yonsei University, Seoul 03722, Korea
* Correspondence: bjteoh@yonsei.ac.kr

**Abstract:** The rapid growth of smartphone financial services raises the need for secure mobile authentication. Continuous authentication is a user-friendly way to strengthen the security of smartphones by implicitly monitoring a user's identity through sessions. Mobile continuous authentication can be viewed as an anomaly detection problem in which models discriminate between one genuine user and the rest of the impostors (anomalies). In practice, complete impostor profiles are hardly available due to the time and monetary cost, while leveraging genuine data alone yields poorly generalized models due to the lack of knowledge about impostors. To address this challenge, we recast continuous mobile authentication as a few-shot anomaly detection problem, aiming to enhance the inference robustness of unseen impostors by using partial knowledge of available impostor profiles. Specifically, we propose a novel deep learning-based model, namely a local feature pooling-based temporal convolution network (LFP-TCN), which directly models raw sequential mobile data, aggregating global and local feature information. In addition, we introduce a random pattern mixing augmentation to generate class-unconstrained impostor data for the training. The augmented pool enables characterizing various impostor patterns from limited impostor data. Finally, we demonstrate practical continuous authentication using score-level fusion, which prevents long-term dependency or increased model complexity due to extended re-authentication time. Experiments on two public benchmark datasets show the effectiveness of our method and its state-of-the-art performance.

**Keywords:** continuous authentication; touch-gesture biometrics; few-shot anomaly detection; data augmentation; temporal convolution network



## 1. Introduction

With the growing use of financial services in smartphones, robust security is required to protect mobile devices from unauthorized access. Nowadays, one-time or static authentication mechanisms such as PIN-, pattern-based authentication, and biometrics, e.g., TouchID, and FaceID, are widely deployed on smartphones. However, they authenticate users only once at the initial log-in and are vulnerable to shoulder-surfing or presentation attacks [1].

Continuous authentication is a promising way to provide an additional security layer for mobile devices, which monitors user identities implicitly through their behavioral biometrics using device built-in sensor data [2–15].

Unlike physical biometrics, which utilizes human physical attributes such as face and fingerprint, behavioral biometrics exploits user behavioral patterns such as screen-touch gestures and gait patterns. Physical biometrics are generally more unique and stable than behavioral ones; however, they require users' cooperation to capture their biometric traits, such as touching a sensor with a finger or turning a face to a camera. In contrast, behavioral biometrics can capture users' biometric data tacitly with built-in sensors of mobile devices such as the touch screen, accelerometer, and gyroscope. Therefore, behavioral biometrics can actualize implicit continuous authentication, further enhancing the security of mobile devices without deteriorating the user experience.

For deployment, mobile continuous authentication systems first capture sensor data of a genuine user (and impostors if available), which is used to construct the user's pro-

file. Then, claimants are monitored during re-authentication time and judged as "accept" or "reject".

*Motivation and Contribution*

Since mobile devices are personal and hardly shared, mobile-based biometric authentication can be deemed an anomaly detection problem, in which models differentiate a single genuine user and all other impostors (anomalies). Prior works have different assumptions about the availability of impostor data with respect to unsupervised, semi-supervised, and fully-supervised learning. The approach relying on unsupervised or semi-supervised anomaly detection assumes that impostor data cannot be obtained in advance, solely depending on unlabeled or genuine data for the training [2–7]. However, this approach is poor at detecting impostors because they do not leverage prior knowledge of impostors.

On the other hand, the fully-supervised approach assumes that genuine and all impostor profiles (seen impostors) are available for learning [3,8–10,16,17]. An apparent issue of this approach is that models are not designed to work in an open-set setting—a practical scenario for biometrics, where it is unfeasible to have all impostor profiles available for training due to time and monetary costs. During authentication, the model may not infer correct decisions when encountering a never-seen impostor (unseen impostor).

Considering the issues of previous approaches, we posit the mobile-based authentication as a few-shot anomaly detection (FSAD) problem [18]. FSAD is a subfield of weakly supervised learning in that only an incomplete set of anomaly classes is known at the training. In addition to the unavailability of complete knowledge of anomalies, FSAD is allowed to use only a few anomalous data. Recent works have addressed FSAD considering the availability of small-scale labeled anomaly data in real-world scenarios [19–22].

Compared to the fully supervised approach, FSAD is practical in mobile-based authentication. First, our model is sample-efficient because it uses only a few *seen impostor* (anomaly class) data for training. Secondly, our model aims to work in an open-set setting, where models encounter *unseen impostors*. By leveraging partial knowledge of seen impostors, our model excels at detecting unseen impostors more than unsupervised and semi-supervised models.

This work is the first attempt to address mobile-based biometrics authentication from the FSAD vein. Specifically, we define the challenges as follows:

1. Limited impostor profiles for training:
   The core challenge of FSAD is the limited availability of seen impostor profiles. However, the models need to be able to detect arbitrary unseen impostors during inference.
2. Class imbalance between genuine and impostor data:
   Since FSAD uses extensive genuine and small impostor data, models inherently suffer from the class-imbalance problem. This can emphasize the genuine class and under-fit the impostor classes, preventing learning the impostor patterns adequately.
3. Large intra-class variance of user behavior data:
   Due to the significant intra-class variance of raw touch sensor data, most touch-gesture authentication solutions have adopted hand-crafted features and conventional machine learning models such as support-vector machine, random forest, etc., which are more noise-robust than learned features [11–14,23–28]. However, hand-crafted features lack refined feature information, making it hard to characterize delicate impostor patterns.

To address the first two challenges, we propose random pattern mixing—a data augmentation method tailored for sequential touch sensor data. While data augmentation has been extensively studied for visual data [29], it is under-explored for anomaly detection [18] and sequential data [17]. A few primitive augmentation methods for sequential data, such as jittering and scaling [3,16,17], are intended to generate data of a minority class (seen impostors in our case) by adding small noises or changing their magnitudes slightly, for example. Despite the fact that smooth decision boundaries between a genuine user and

seen impostors yield better generalization, they do not help detect unseen impostors because unseen impostors may have completely different patterns from seen impostors. To improve the robustness against unseen impostors, the proposed random pattern mixing aims to enlarge the feature space of impostor data, producing various synthetic patterns not present in seen impostors.

To respond to the third challenge, we devise a novel local feature pooling (LFP)-enabled temporal convolution network (TCN) for raw sequential touch sensor data modeling. The TCN [30] is a variant of the convolution neural network (CNN) modified for sequential data modeling. Kim et al. exploited the TCN for touch-gesture authentication, showing its superior performance over the LSTM baseline [31]. Indeed, the TCN can learn global information with its large receptive field and yields noise-robust global representations of touch screen data.

In this paper, we enhance the TCN with the LFP module, which enables aggregating local feature information to learn fine-grained representations of impostors. Specifically, the LFP module prevents local information loss due to the sparsity of the kernel in the dilated convolution. Moreover, the LFP module is parameter-free; thus, it is suitable for applications on memory-constrained mobile devices such as smartphones.

Finally, we demonstrate continuous authentication based on the decision and score-level fusion. Although many researchers have worked on continuous authentication, prior works have applied the sensor-level fusion, which enlarges window size according to re-authentication time [2,3,5,10]. However, this approach exaggerates the long-term dependency problem for the recurrent neural networks (RNNs) or increases the model complexity of the CNNs. Therefore, we explore alternatives using the decision and score-level fusion, which maintain the window size and prevent the problems above.

In summary, our contributions are as follows.

- We formulate continuous mobile-based biometric authentication as a few-shot anomaly detection problem, aiming to enhance the discrimination robustness of the genuine user and seen and unseen impostors.
- We present a sequential data augmentation method, random pattern mixing, which expands the feature space of impostor data, providing synthetic impostor patterns not present in seen impostors.
- We propose the LFP-TCN, which aggregates global and local feature information, characterizing fine-grained impostor patterns from noisy sensor data.
- We demonstrate the continuous authentication based on the decision and score-level fusion, which can effectively improve the authentication performance for longer re-authentication time.

The rest of this paper is organized as follows. Section 2 reviews related works. Section 3 introduces our authentication method. Section 4 explains the experimental settings. Section 5 presents the experimental results and analysis. Finally, we end with the conclusion in Section 6.

## 2. Related Works

This section presents relevant works on touch-gesture features, anomaly detection in mobile-based biometrics, few-shot anomaly detection, and data augmentation for mobile-based biometrics.

### 2.1. Touch-Gesture Features for Authentication

Touch-gesture authentication verifies users based on screen-touch behaviors, summarized by touch trajectory, device motion, and orientation. Prior works either utilize the screen touch coordinates, pressure, or motion sensor data captured by the accelerometer and gyroscope [10–15,24–28].

Most existing methods have relied on hand-crafted features such as length of trajectory, the median of finger pressure, etc. [11–15,24–28]. This is mainly attributed to touch-gesture being sensitive to behavioral changes. Typically, the systems first extract

hand-crafted features and train prediction models, such as the one class-SVM [12,15], kernel ridge regression [26], random forest [27], temporal regression forest [25], SVM [11,13], and MLP [14,24,28]. Criticisms for hand-crafted features are that they are task- or user-dependent and require expert knowledge to design, which are not trivial. Furthermore, hand-crafted features lack delicate features and are unlikely to characterize refined impostor patterns.

Several works attempted to combine hand-crafted and deep learned features [23] or build an end-to-end feature learning model [10,31]. To address the limitations of hand-crafted features, we opt to learn the features from raw touch-gesture data directly with a deep network.

### 2.2. Deep Anomaly Detection in Mobile-Based Biometrics

Mobile-based biometric authentication is often formulated as an anomaly detection problem. Here, we limit the scope to deep learning-based anomaly detection models. Based on how a model is trained, the existing models can be broadly categorized into unsupervised, semi-supervised, or fully-supervised ones. Table 1 tabulates several recent deep anomaly detection models in mobile-based biometrics. Most methods utilize motion sensor data, such as the accelerometer, gyroscope, and magnetometer. Many works utilize behavior data while walking [3,9,16] or operating a smartphone [2,4–8,10,17].

**Table 1.** End-to-end models for mobile-based biometrics.

|  | Sensor | Supervision | Architecture |
|---|---|---|---|
| Centeno et al. [2] | Acc. [1] | Unsupervised | Autoencoder |
| Giorgi et al. [3] | Acc. [1], Gyro. [2] | Unsupervised | LSTM autoencoder |
| CNNAuth [4] | Acc. [1], Gyro. [2] | Semi-supervised | CNN + PCA [5] + OC-SVM |
| SCANet [5] | Acc. [1], Gyro. [2] | Semi-supervised | CNN + PCA [5] + OC-SVM |
| DeFFusion [6] | Acc. [1], Gyro. [2] | Semi-supervised | CNN + FA [6] + OC-SVM |
| CAuSe [7] | Acc. [1], Gyro. [2], Mag. [3] | Semi-supervised | CNN + PCA [5] + LOF |
| Mekruksavanich et al. [16] | Acc.[1], | Fully supervised | CNN |
| Giorgi et al. [3] | Acc. [1], Gyro. [2] | Fully supervised | LSTM |
| DeepAuth [8] | Acc. [1], Gyro. [2] | Fully supervised | LSTM |
| Benegui et al. [17] | Acc. [1], Gyro. [2] | Fully supervised | LSTM/ConvLSTM |
| DeepAuthen [9] | Acc. [1], Gyro. [2], Mag. [3] | Fully supervised | ConvLSTM |
| AUTOSen [10] | Touch. [4], Acc. [1], Gyro. [2], Mag. [3] | Fully supervised | Bidirectional LSTM |

[1] Accelerometer, [2] Gyroscope, [3] Magnetometer, [4] Touch screen, [5] Principal component analysis, [6] Factor analysis.

Autoencoder is a popular architecture for unsupervised learning [2,3]. This approach uses an encoder network for input dimension reduction and a decoder network to reconstruct inputs and compute reconstruction losses as anomaly scores.

The semi-supervised models are composed of a pretrained CNN feature extractor, dimensionality reduction, and a one-class classifier such as OC-SVM [4–7]. Many of them are designed as light-weighted models, accounting for the deployment in mobile devices [4–6]. An advantage of this approach is that classifiers can leverage powerful representations from pretrained feature extractors. However, the training of the feature extractors is completely decoupled from that of the classifiers. Therefore, the learned representations are not optimized for the classifiers. Besides, the disuse of impostor data yields decision boundaries without characterizing impostor patterns, often failing to detect impostors when genuine or unlabeled data have large intra-class variances.

The other approach is fully supervised anomaly detection, in which a large set of genuine and labeled impostor data are given at the training. Existing works have mainly utilized CNNs [16], LSTM [3,8], and ConvLSTM [9,17]. Using the large-scale labeled impostor data, they reduce an anomaly detection problem to a binary (a genuine user with a positive label vs. the rest with a negative label) or multi-class (a genuine user vs. multiple impostors) classification problem. The use of impostor data can significantly boost the

model performance [3]. However, the availability of large impostor data is often unrealistic due to time and monetary costs. In particular, it is not feasible to utilize full knowledge of impostors because they can be unseen in biometrics.

One related approach to ours (FSAD) is weakly-supervised anomaly detection. It aims to detect unseen impostors by utilizing labeled genuine and seen impostor data for the training. Kim et al. explored this approach in touch-gesture authentication [31]. However, large-scale impostor data is necessary for weakly supervised models, which have the same constraint as the fully supervised approach.

In this work, we study few-shot anomaly detection (FSAD), which addresses the drawbacks of previous approaches by leveraging small-scale seen impostor data.

### 2.3. Few-Shot Anomaly Detection

Few-shot anomaly detection is a recently emerged problem, which considers the availability of a few labeled anomaly data for training [18]. Pang et al. proposed the deviation loss for anomaly image detection problems, which forces anomaly scores of anomalies to be statistically significant [19]. Ding et al. leveraged a meta-learning technique to transfer knowledge of anomalies from multiple few-shot anomaly detection tasks [20]. Finally, Tian et al. trained an encoder, maximizing the mutual information between normal images and their embedding, and utilized the deviation loss to optimize a score inference network [21]. However, none of the works directly addressed the class imbalance in FSAD, which may cause under-fitting to the minority class (impostor data).

In this work, we explore the FSAD solution for anomaly touch-gesture detection problems. Our paper is the first to do so to the best of our knowledge.

### 2.4. Data Augmentation for Mobile-Based Biometrics

Data augmentation is a common approach to address the class-imbalance problem [29]. Typical augmentation methods for time series data are categorized into random transformation, pattern mixing, and generative model-based methods [32].

The random transformation-based methods generate new time series using some transformation function on a minority class data. Representative methods are jittering, scaling, permutation, warping, rotation, cropping, etc., commonly used in mobile-based biometrics (Table 2). The authors of [3,7,15–17] applied jittering, scaling, permutation, etc. for motion sensor data.

**Table 2.** Data augmentation methods applied in mobile-based biometrics.

| | Sensor | Feature | Data Augmentation Methods |
|---|---|---|---|
| Agrawal et al. [11] | Touch. [1] | Hand-crafted | GAN generator |
| SWIPEGAN [27] | Touch. [1], Acc. [2] | Hand-crafted | GAN generator |
| DAKOTA [13] | Touch. [1], Acc. [2], Gyro. [3], Mag. [4] | Hand-crafted | SMOTE |
| SensorAuth [15] | Acc. [2], Gyro. [3] | Hand-crafted | Jittering, scaling, permutation, sampling, cropping |
| Mekruksavanich et al. [16] | Acc. [2] | Raw | Jittering, scaling, warping, rotation |
| Benegui et al. [17] | Acc. [2], Gyro. [3] | Raw | Jittering, scaling, warping |
| Giorgi et al. [3] | Acc. [2], Gyro. [3] | Raw | Jittering, scaling, permutation |
| CAuSe [7] | Acc. [2], Gyro. [3], Mag. [4] | Raw | Jittering, scaling, permutation, cropping, warping, rotation |

[1] Touch screen, [2] accelerometer, [3] gyroscope, [4] magnetometer.

The pattern mixing-based methods produce a new time series by combining multiple time series. Interpolation and the SMOTE [33] are popular methods. The SMOTE interpolates a minority class sample and its *k*-nearest sample. Incel et al. applied this method for touch screen and motion sensor data [13]. Our proposed random pattern mixing belongs to this category; however, it differs because existing methods combine class-constrained seen patterns, while ours does class-unconstrained random patterns.

Several researchers attempted the generative model-based methods. For example, the authors in [11,27] utilized a GAN for hand-crafted touch screen data. However, this approach demands a large set of training data from the minority class, which is unavailable in FSAD as we use only a few impostor data.

## 3. Method

This section first gives an overview of our proposed authentication system and then introduces each component, including the random pattern mixing, LFP-TCN, and decision-making mechanism.

### 3.1. System Overview

Figure 1 illustrates an overview of our proposed FSAD mobile-based continuous authentication system. For data acquisition, the system collects the touch screen and motion sensor data from the accelerator and gyroscope when a user touches a smartphone screen.

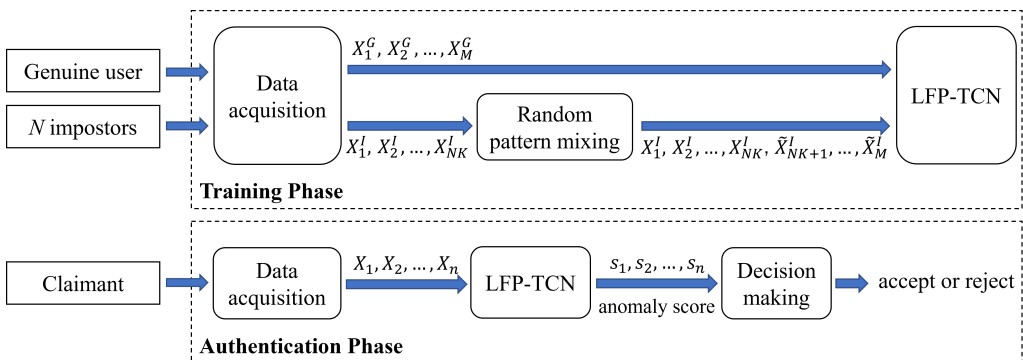

**Figure 1.** Framework of our authentication method.

These sensors capture 11 raw touch-gesture readings with 100 Hz. The touch screen gives x-y coordinates of touched points, touch pressure, and horizontal-vertical touched area information. The accelerometer and gyroscope give x-y-z axis values of velocity and rotation, respectively. Each sensor reading is min-max normalized.

The captured sensor readings are then converted into data samples by the sliding window with the window size of $T_w$ seconds ($100T_w$ readings) and overlapping of $T_o$ seconds. Each raw reading is concatenated along the time dimension. We obtain a raw data matrix $X = \{f_t, f_a, f_g\}^\top \in \mathbb{R}^{11 \times 100T_w}$, where $f_t$, $f_a$, and $f_g$ are the raw readings from the touch screen, accelerometer, and gyroscope, respectively.

In the training phase, the system utilizes $M$ genuine samples, $X_1^G, X_2^G, \ldots, X_M^G$, and $K(=10)$ samples from each of $N(=10)$ impostors, $X_1^I, X_2^I, \ldots, X_{NK}^I$. After that, the random pattern mixing augments the impostor samples to $M$ samples, $X_1^I, X_2^I, \ldots, X_{NK}^I, \tilde{X}_{NK+1}^I, \ldots, \tilde{X}_M^I$, so that the number of genuine and impostor samples is balanced up. Finally, they are used to train the LFP-TCN with the binary cross entropy loss.

In the authentication phase, the sensor data are continuously captured, and a claimant is verified every $T_r$ seconds for re-authentication. Here, $n = 1 + \frac{T_r - T_w}{T_w - T_o}$ samples can be acquired by the sliding window for each verification. For instance, we obtain five samples when $T_r = 3$, $T_w = 1$, and $T_o = 0.5$. For $n$ samples, the LFP-TCN returns $n$ anomaly scores, $s_1, s_2, \ldots, s_n$. Finally, the decision-making module yields a decision of "accept" or "reject" based on the decision or score-level fusion of the $n$ scores.

### 3.2. Random Pattern Mixing

To better illustrate our proposed augmentation method, we first revisit augmentation techniques designed for time series, namely jittering, scaling, permutation, and SMOTE [33].

Suppose $X = \{S_1, S_2, \ldots, S_d\}^\top \in \mathbb{R}^{d \times T}$ is a multivariate time series that has d(=11) variate, and each variate $S_i \in \mathbb{R}^T$ has a length of $T(=100T_w)$.

### 3.2.1. Jittering

Jittering is one of the most common augmentation techniques for adding noise to an original signal. In this work, Gaussian noises are adopted:

$$\tilde{S}_i = S_i + \{\epsilon_1, \epsilon_2, \ldots, \epsilon_T\}, \forall \epsilon_j \sim \mathcal{N}(0, \sigma^2) \tag{1}$$

where $\sigma$ is a hyper-parameter. Then, a generated sample is:

$$\tilde{X} = \{\tilde{S}_1, \tilde{S}_2, \ldots, \tilde{S}_d\}^\top \tag{2}$$

### 3.2.2. Scaling

Scaling is another popular augmentation technique, which changes the magnitude of a time series according to a scaling factor. We take the scaling factor from the Gaussian distribution:

$$\tilde{S}_i = S_i \odot \{\epsilon_1, \epsilon_2, \ldots, \epsilon_T\}, \forall \epsilon_j \sim \mathcal{N}(1, \sigma^2) \tag{3}$$

where $\odot$ denotes the element-wise product, and $\sigma$ is a hyper-parameter. A new sample is given by following Equation (2).

### 3.2.3. Permutation

Permutation rearranges segments of a time series in order to produce a new pattern. Permutation splits a time series in the time axis into $P$ segments with a length of $\frac{T}{P}$ and permutes them.

### 3.2.4. SMOTE

The SMOTE [33] is a popular augmentation method for imbalanced data. It randomly selects a sample $X$ from a minority class and its $k$-nearest sample $X_{NN}$. Then, a synthetic sample is produced by interpolating these samples:

$$\tilde{X} = X + \lambda|X - X_{NN}| \tag{4}$$

where $\lambda$ is a random value in [0, 1].

These conventional augmentation methods generate synthetic samples of a minority class (seen impostors in our case) within the feature space of seen classes. However, unseen impostors have unbounded and arbitrary behavior patterns. Therefore, augmenting impostor data in class-constrained feature space may not sufficiently increase the generalizability against unseen impostors.

**Random Pattern Mixing**

To improve the robustness to unseen impostors, we propose random pattern mixing. We hypothesize that the feature space of unseen impostors is larger than that of seen impostors. This hypothesis is reasonable in our task because seen impostors are only partial profiles of a few impostors, while unseen impostor profiles are unbounded. To simulate unseen impostor features, we enlarge the feature space of impostors by mixing seen impostor and uniform random patterns. Since random patterns are class-unconstrained, our method can produce synthetic instances not present in the feature space of seen impostors.

Given a matrix $R = \{U_1, U_2, \ldots, U_d\}^\top \in \mathbb{R}^{d \times T}$, in which each value in $U_i \in \mathbb{R}^T$ follows the uniform distribution:

$$U_i = \{\epsilon_1, \epsilon_2, \ldots, \epsilon_T\}, \forall \epsilon_j \sim \mathcal{U}(0, 1) \tag{5}$$

where the uniform distribution is bounded by [0, 1], supposing every sample is min-max normalized.

Then, we mix the seen impostor and random patterns. By letting a seen impostor sample be $X$, it follows:

$$\tilde{X} = M \odot X + (1 - M) \odot R \tag{6}$$

where $\odot$ is the element-wise product, and $M \in \mathbb{R}^{d \times T}$ is a mask of zeros and ones, in which the ratio of the number zero is adjustable. As the mask contains fewer zeros, the feature space of the generated patterns becomes as narrow as seen impostors. When the values in the mask are all ones, our method is equivalent to up-sampling.

In a nutshell, the proposed random pattern mixing can efficiently enlarge the feature space of impostors, diversifying impostor features in training data.

### 3.3. Local Feature Pooling-Based Temporal Convolution Network (LFP-TCN)

The TCN is a CNN variant for sequential data modeling [30]. The characteristics of TCN are manifested by causal convolution and dilated convolution. In Figure 2, the blue lines illustrate convolutional filters by the dilated causal convolution with the kernel size of 3. The causal convolution is performed on elements only at the same and earlier time steps in the previous layer. This is achieved by the causal padding, i.e., adding zero-padding of length of (kernel size × dilation rate − 1) in front of inputs. Additionally, the dilated convolution is performed on non-successive elements with a fixed step (dilation rate), which efficiently enlarges the receptive field. These characteristics enable the TCN to learn relationships of very long sequences effectively.

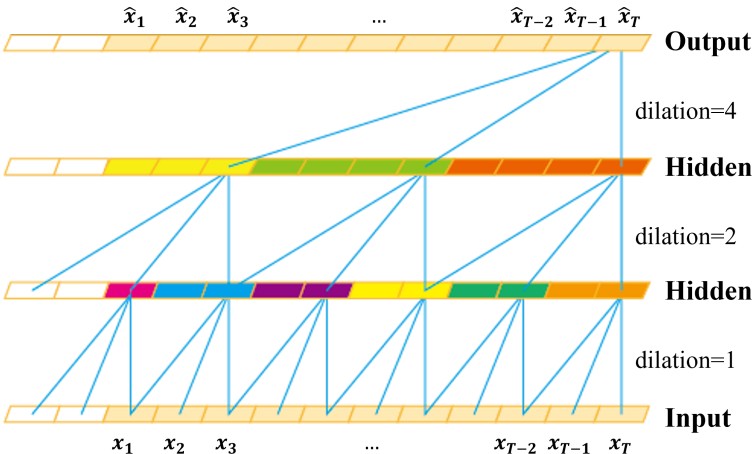

**Figure 2.** Convolution by the LFP-TCN: The convolution is performed with the kernel size of 3 and dilation rate of $(1, 2, 4)$. The colored elements in hidden layers illustrate max-pooled elements. The kernel size of the max-pooling is the same as the dilation rate on each layer. This operation compresses local information of neighboring non-filtered elements.

On the other hand, the dilated convolution fails to aggregate local feature information neighboring elements have [34,35]. This is because the dilated convolution kernel is spatially discrete. In image segmentation, this problem hinders the learning of small objects. Hamaguchi et al. addressed it by adding the local feature extraction (LFE) module, composed of convolutional layers with decreasing dilation rates along with the depth of layers [34]. Park et al. proposed the concentrated-comprehensive convolution, in which two depth-wise convolutions are performed before the dilated convolution [35].

The local information loss also occurs in the TCN. For example, in Figure 2, the output at the last time step $\hat{x}_T$ is affected by the elements connected with blue lines. We can observe that every detail is connected between layers when the dilation rate is one. In contrast, the connection becomes more sparse as the dilation rate increases, and many hidden layer elements are completely non-connected. This means that more feature information is lost in higher layers, and only specific global features are aggregated at the output layer. Note that classifiers for time series are based on the many-to-one architecture, classifying inputs

based on the output only at the last time step. This issue forces the TCN to focus on specific global information and prevents learning refined representations of touch-gestures.

However, for this problem, the existing methods [34,35] are not preferable in mobile-based biometrics because using additional convolution layers increases model complexity. Considering the limited memory capacity of mobile devices, we designed the LFP module, a parameter-free method to prevent local information loss. Specifically, the LFP module extracts salient feature information from neighboring elements with max-pooling before the dilated convolution is performed.

Figure 2 illustrates the convolution by the LFP-TCN. The LFP module performs the causal padding and max-pooling to inputs of convolution layers when their dilation rates are more than one. The causal padding is used to maintain the time consistency and input length. Additionally, max-pooling is used to compress features of the non-filtered elements in a skipped interval with the length of a dilation rate (the same color elements in Figure 2). To this end, the kernel size of the pooling layer is set as the dilation rate of the following convolution layer.

Figure 3a depicts the base unit of LFP-TCN. The LFP module first applies the causal padding and then max-pools every non-filtered element in the skipped interval. By doing this, all elements in the interval are max-pooled first, and the compressed feature information is convoluted, propagating local feature information into higher layers.

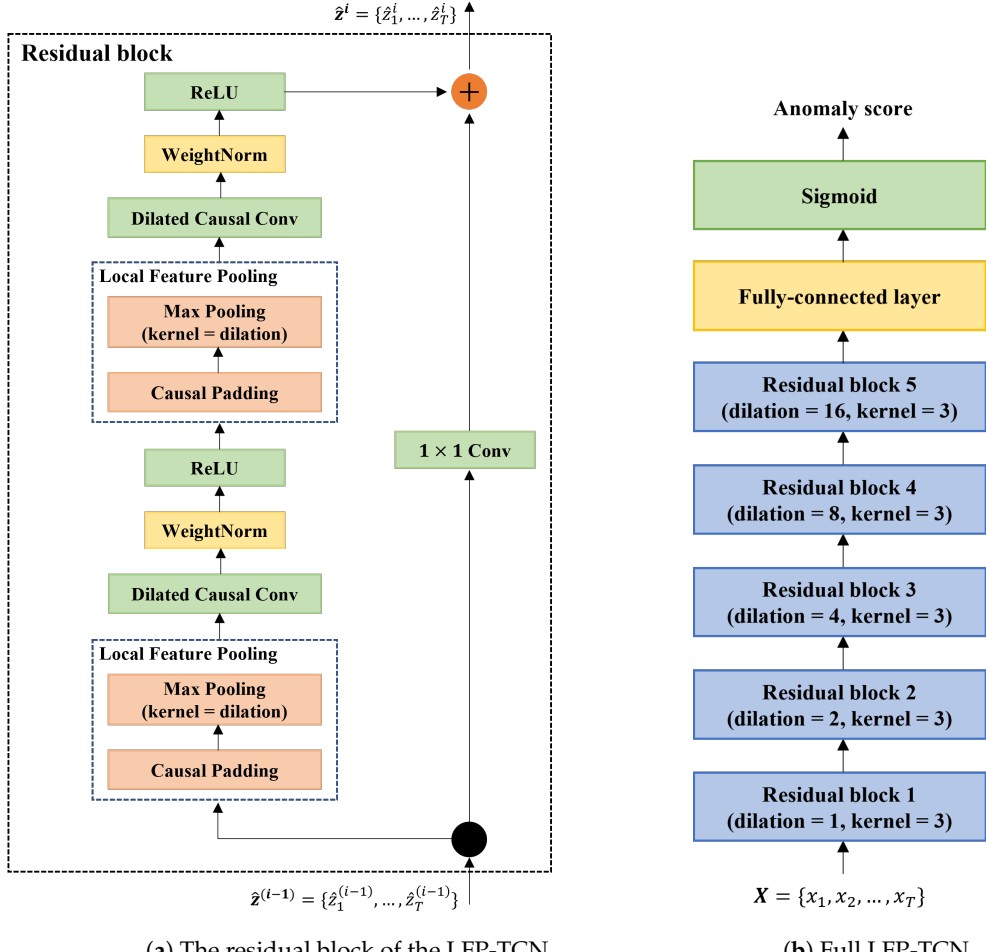

(a) The residual block of the LFP-TCN        (b) Full LFP-TCN

**Figure 3.** (a) The LFP modules are added in front of the dilated causal convolution layers. They aggregate local information of inputs and pass compressed feature information to the convolution layers. (b) Full LFP-TCN consists of 5 stacked residual blocks and a fully connected layer.

Figure 4 illustrates a toy example of the LFP module. In the example, the sequence of $(6, 1, 5, 3, 4, 2)$ is convoluted with a filter kernel of $(1, 2, 3)$ and a dilation rate of two. Without

the LFP module, the elements of $(1, 3, 2)$ are convoluted, where the information of the non-filtered elements is wholly discarded. On the contrary, the LFP module causally max-pools the sequence with the kernel size of two, and the elements of $(6, 5, 4)$ are convoluted. As a result, more salient features of non-filtered elements are propagated into the next layer, helping learn fine-grained representations of impostors.

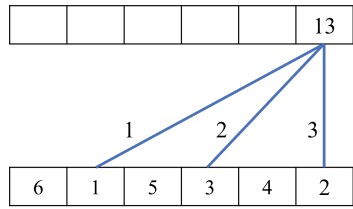 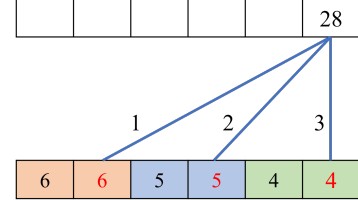

Dilated causal convolution   LFP module + Dilated causal convolution

**Figure 4.** Toy example of the LFP module: the sequence of $(6, 1, 5, 3, 4, 2)$ is convoluted with and without the LFP module. The dilated causal convolution has a filter kernel of $(1, 2, 3)$ and a dilation rate of two. The LFP module extracts salient features from neighboring elements in the dilation interval.

Our LFP-TCN comprises 5 stacked residual blocks, as illustrated in Figure 3b. The dilation rate is set as $(1, 2, 4, 8, 16)$ for each block, and all convolution layers contain 128 kernel filters with size 3. The outputs at the last time step are fed into the output layer, a fully connected layer with a single neuron, to render an anomaly score. The higher anomaly scores indicate that the claimant is more likely to be an impostor. This model is trained with the binary cross-entropy loss.

### 3.4. Decision-Making Mechanism

Continuous authentication yields a decision of "accept" or "reject" after every re-authentication. In general, longer re-authentication time improves authentication performance due to better-accumulated information; however, few works have discussed how to make decisions with variable re-authentication time.

Primary decision-making mechanisms are categorized into the sensor, decision, and score-level fusion [36]. Some prior works have followed the sensor-level fusion, enlarging the window size (input sequence length) according to re-authentication time ($T_w = T_r$) [2,3,5,10].

However, due to the long-term dependency, networks such as RNN may not fully benefit from longer input sequences. Moreover, a larger window size increases the model size of CNNs, increasing the output feature map size [5]. Additionally, the TCN requires more layers or a larger kernel size to have a larger receptive field to cover input sequences. Therefore, sensor-level fusion is problematic regarding performance stability and model complexity.

We consider the decision and score-level fusion as alternatives to the sensor-level fusion, which maintains the window size and is free from the above problems.

### 3.4.1. Decision-Level Fusion

The decision-level fusion combines decisions from multiple classifiers [36]. However, in mobile continuous authentication, it is not feasible to utilize many different classifiers because of the limited memory capacity of mobile devices. Therefore, we resort to accumulating a number of decisions with a single model by dividing the captured data into multiple segments. To this end, the data acquisition in Figure 1 applies the sliding window with a window size shorter than re-authentication time ($T_w < T_r$).

The entire data acquisition renders $n$ data samples. Then, the obtained samples deliver $n$ decisions of "accept" or "reject" based on a preset threshold. Finally, we can obtain a final decision based on the majority voting on these decisions.

### 3.4.2. Score-Level Fusion

The score-level fusion combines multiple scores from multiple classifiers. Popular approaches are taking the mean or maximum value of multiple scores [36]. Similar to decision-level fusion, we consider taking multiple scores with a single model by dividing the captured data into multiple segments.

With $n$ samples obtained in re-authentication time, the model renders $n$ anomaly scores; then, we can compute the combined scores by fusion. This work takes the mean and maximum of the obtained scores.

Since the decision and score-level fusion does not increase the window size with respect to re-authentication time, the RNNs and CNNs do not suffer from the long-term dependency problem or increase the model complexity.

## 4. Experiments

This section describes our experiment settings, including datasets, experiment protocol, model hyperparameters, and accuracy metrics.

### 4.1. Dataset

Two publicly available datasets were used for our experiments. We utilized three types of sensor data, i.e., touch screen, accelerometer, and gyroscope data, while users touch a smartphone screen.

- The HMOG dataset [37] is a multimodal dataset containing touch screen, accelerometer, and gyroscope data captured from 100 subjects while doing three tasks, e.g., reading web articles, typing texts, and navigating a map. The dataset comprises 24 sessions (8 sessions for each task) for each subject. The data are captured under sitting and walking situations. We used data of these tasks under sitting situations.
- The BBMAS dataset [38] captures touch screen, accelerometer, and gyroscope data from 117 subjects while typing free and fixed texts. These data are captured under sitting and walking situations. We used only data under the sitting situations. We removed five subjects' data because of too small a data size, resulting in 112 subjects' data.

In both datasets, we obtained data samples by a sliding window with a window size $T_w$ of 1 s (100 sensor readings). The overlapping of the window $T_o$ was set as 0.5 s (50 readings) in the HMOG dataset and 0.9 s (90 readings) in the BBMAS dataset. We obtained an average of 1672 and 862 samples per user in the HMOG and BBMAS datasets, respectively.

### 4.2. Experiment Protocol

To prepare for seen and unseen impostors, we divided the subjects randomly in half in each dataset. The first half (50 subjects in the HMOG dataset and 56 subjects in the BBMAS dataset) was used for training, i.e., a genuine user and seen impostors. We regarded one of the subjects as a genuine user and $N(=10)$ subjects as seen impostors. All subjects in the second half were used for the testing as unseen impostors. Then, we repeated this procedure, using the second half for training and the first half for unseen impostors.

Considering the real-world usage of continuous authentication, we maintained the temporal dependency of the training and test data: the initial 80% of genuine samples were taken for the training, and the latter 20% of them were for testing. Similarly, initial consecutive $K(=10)$ samples were taken from each of $N(=10)$ seen impostors, and the rest were used for testing. To adhere to the few-shot protocol, we extracted only ten samples from each of the ten impostors, i.e., $N = K = 10$. As the choice of seen impostors can be an essential factor for the performance, we fixed seen impostors for each genuine user across whole experiments.

For the testing, impostor data amounted to 3000 samples, in which 1500 samples were randomly taken from unseen impostors. Finally, considering the randomness of the results,

we repeated the training and testing five times and reported the average performance over the five trials.

### 4.3. Implementation Detail

If data augmentation was applied, we used the binary cross entropy loss. Otherwise, we used the balanced binary cross-entropy loss, whose class-weighting parameter is adjusted according to the ratio of impostor samples in the training data:

$$\mathcal{L} = -y\log(p_t) - \lambda(1-y)\log(p_t) \tag{7}$$

where $y$ is the ground-truth label, $p_t$ is the target probability, and $\lambda$ is the class-weighting parameter. We set $\lambda$ as the ratio of genuine samples to impostor ones in the training set.

Our models were trained with the Adam optimizer and batch size of 64. In the HMOG dataset, the number of epochs was 100, and the learning rate was $1 \times 10^{-3}$.

For experiments on the BBMAS dataset, models were pretrained on the HMOG dataset and fine-tuned with the BBMAS dataset for 50 epochs. The learning rate was $1 \times 10^{-4}$ for the residual blocks and $1 \times 10^{-3}$ for the output layer.

### 4.4. Accuracy Metrics

We evaluate model performance with the false rejection rate (FRR), the false acceptance rate (FAR), and the equal error rate (EER). The FRR represents the ratio at which a genuine subject is falsely rejected. Similarly, the FAR is the ratio at which impostors are falsely accepted. The EER is a point where the FRR = FAR, depending on a threshold. In the experiments, we present the averaged EER over each user's model, e.g., 100 models in the HMOG dataset.

## 5. Experimental Result

### 5.1. Effectiveness of the LFP Module

We compare the LFP-TCN with baselines, including the MLP, LSTM, CNN, and TCN. The output layer for each model is a fully-connected layer with a single neuron, which renders anomaly scores. These baselines have almost the same parameter size as our LFP-TCN. Moreover, we compare our LFP module with the LFE module [34] to handle the local information loss. The details of each model are as follows:

- MLP: 3-layer MLP with neurons of $(400, 200, 128)$ on each layer. We use the ReLU activation function.
- LSTM: 5 stacked LSTM blocks with hidden states of 128. The hidden states at the last time step are fed into the output layer.
- CNN: 10 convolution layers with a kernel size of three. ReLU is used for the activation function. The final feature map is flattened and fed into the output layer. This is equivalent to the below TCN, in which we replace the dilated causal convolution layers with ordinary convolution layers and remove the skip connection.
- TCN: 5 stacked residual blocks in Figure 3 without the LFP module.
- LFE-TCN: the above TCN with the LFE module [34]. The LFE module puts an additional five residual blocks on top of the TCN with the decreasing dilation rate, resulting in 10 stacked residual blocks with the dilation rate $(1, 2, 4, 8, 16, 16, 8, 4, 2, 1)$ in each block.

Table 3 demonstrates the averaged EERs and the number of parameters for different models. As depicted in Table 3, the TCN outperforms the MLP, LSTM, and CNN with an EER of 12.76% and 14.02% in the HMOG and BBMAS dataset, respectively. The TCN can learn noise-robust features from sensor data, exploiting its sizeable receptive field.

On top of that, our LFP-TCN achieved the lowest EERs in both datasets, showing values of 12.25% and 13.70%, respectively. This result shows the effectiveness of our LFP module, which improved the EER by 0.51% and 0.32% in each dataset. This improvement

is contributed by aggregating local feature information, yielding fine-grained impostor representations.

Additionally, we implemented the LFE module with the TCN, even though it was originally designed for image segmentation. We confirmed that the LFE module doubled the parameters and did not improve the EER in our task. On the other hand, our LFP module kept the parameter size, and the LFP-TCN outperformed the baselines.

**Table 3.** Averaged EERs (%) for different architectures.

| Architecture | Dataset | | Parameter Size |
| --- | --- | --- | --- |
| | HMOG | BBMAS | |
| MLP | 20.33 | 15.48 | 0.5 M |
| LSTM | 23.55 | 14.09 | 0.6 M |
| CNN | 13.14 | 16.97 | 0.4 M |
| TCN | 12.76 | 14.02 | 0.5 M |
| LFE-TCN | 14.90 | 16.76 | 1.0 M |
| LFP-TCN | **12.25** | **13.70** | 0.5 M |

### 5.2. Effectiveness of the Random Pattern Mixing

This experiment evaluates our random pattern mixing. For comparison, we apply jittering, scaling, permutation, up-sampling, and the SMOTE to the LFP-TCN. We tune the hyper-parameters for each method. For the jittering, we add Gaussian noises with the standard deviation $\sigma \in \{0.05, 0.1, 0.15\}$. For the scaling, we set the scaling factor according to Gaussian distribution $\mathcal{N}(1, \sigma)$, where $\sigma \in \{0.05, 0.1, 0.2\}$. For the permutation, we consider the number of segments $P \in \{5, 10, 20\}$. For the SMOTE, we interpolate a sample with its $k$-nearest neighbor where $k \in [1, 9]$ with a step of 2. Finally, for the random pattern mixing, we adjust the ratio of the number zero in the mask (the ratio of injected random patterns) in $\{0.05, 0.1, 0.2, 0.3, 0.4, 0.5\}$.

Figure 5 illustrates the averaged EERs for different data augmentation methods. As illustrated in Figure 5, the random pattern mixing outperforms other methods with the margins of 0.27% and 0.18% in the HMOG and BBMAS datasets, respectively. In particular, the use of random pattern mixing improves the EER by 1.30% and 0.85% over the model without data augmentation in each dataset.

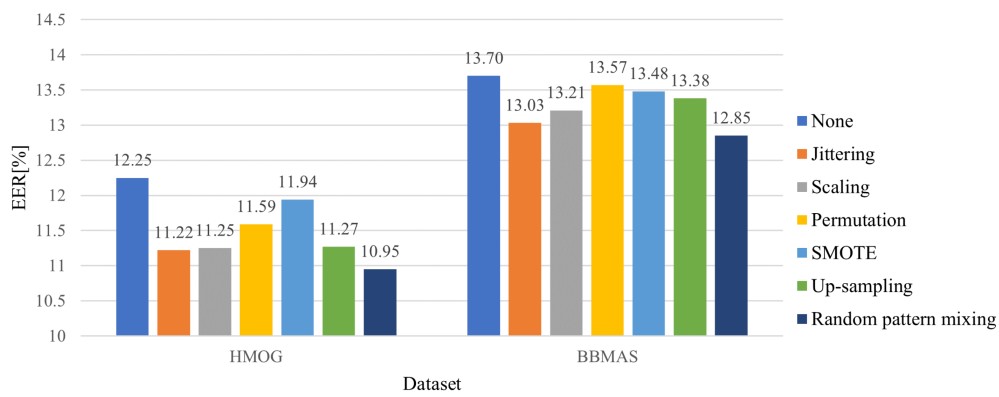

**Figure 5.** Averaged EERs (%) for different data augmentation methods.

Interestingly, the up-sampling improved the EER by 0.98% and 0.32% in each dataset. This is because it prevents the penalties for the impostor class from being heavily biased to a few impostor instances in a batch, which helps balance the genuine and impostor classes.

Figure 6 depicts the averaged EERs against only unseen impostors. We can see that every existing method improves the EERs in both datasets. This is because some unseen impostors share a part of the feature space with seen impostors.

Among competing methods, the random pattern mixing is the most robust to unseen impostors, with the EER of 12.33% and 14.58% in each dataset. Note that the difference between the random pattern mixing and the up-sampling is whether the uniform-based random patterns are injected into seen impostor set. Figure 6 shows that mixing random and seen impostor patterns further reduce the EERs from up-sampling by 0.77% and 0.97% in each dataset. This implies that the random pattern mixing can effectively simulate unseen impostor patterns.

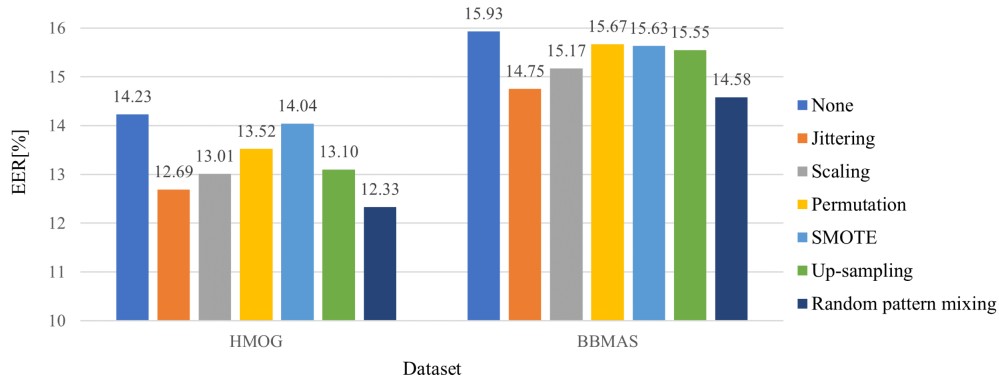

**Figure 6.** Averaged EERs (%) against only unseen impostors for different data augmentation methods.

### 5.3. Comparison with State of the Arts

This experiment compares our method with state-of-the-art techniques in different sets of sensors. First, we pick some recent mobile-based authentication methods based on semi-supervision and full supervision, which comply with the HMOG and BBMAS datasets. Then, we train and test those models with the same procedure of Section 4.2.

We reproduce the models in [10,13,14] for the touch screen, accelerometer, and gyroscope. The AUToSen [10] is an end-to-end learning-based model, while the others are based on hand-crafted features. The DAKOTA [13] first extracts hand-crafted features and verifies users by the SVM. Volaka et al. [14] train a three-layers MLP on hand-crafted features, which has 128 neurons in each layer.

The competing models for accelerometer and gyroscope [3,4,6,9] are all end-to-end models. The CNNAuth [4] and DeFFusion [6] are semi-supervised models, while the others are fully supervised models.

Table 4 shows the averaged EERs of SOTA methods. As illustrated in Table 4, our method achieves the lowest EERs for both sets of sensors. For the touch screen, accelerometer, and gyroscope, our model showed the EER 10.95% and 12.85% in the HMOG and BBMAS datasets, outperforming SOTA methods with margins of 0.76% and 1.21%, respectively. For the accelerometer and gyroscope, the proposed model achieves an EER of 18.32% and 22.98% in each dataset, outperforming others with margins of 8.04% and 0.30%, respectively.

We observe that the non-end-to-end methods, such as the DAKOTA, were superior to the end-to-end AUToSen. This is because deep features are sensitive to the intra-class variance of touch screen data. Even though our method is end-to-end, it significantly outperformed the AUToSen by 11.18% and 2.49% in each dataset. This is because LFP-TCN learns fine-grained representations of impostors from noisy data by aggregating global and local feature information.

Overall, our proposed method is superior to other methods in three aspects. The first is the use of scarce impostor data. Table 4 shows that the impostor-informed models, such as ours and Giorgi et al., significantly outperformed the semi-supervised models for the accelerometer and gyroscope data. Another advantage is model architecture; a few competing models, such as AUToSen and DeFFusion, are based on LSTM and CNN. In addition, the LFP-TCN is a more robust architecture designed to characterize detailed

impostor patterns. Finally, our model learns diverse impostor patterns with random pattern mixing, which generates synthetic impostor instances not present in seen impostors.

**Table 4.** Averaged EERs (%) of SOTA methods.

| Method | Sensor | Feature | Supervision | Dataset | |
|---|---|---|---|---|---|
| | | | | **HMOG** | **BBMAS** |
| AUToSen [10] | | Raw | Fully-supervised | 22.13 | 15.34 |
| DAKOTA [13] | To. [1], Acc. [2], Gyro. [3] | Hand-crafted | Fully supervised | 11.71 | 14.06 |
| Volaka et al. [14] | | Hand-crafted | Fully supervised | 15.59 | 15.04 |
| Ours | | Raw | Weakly supervised | **10.95** | **12.85** |
| CNNAuth [4] | | Raw | Semi-supervised | 31.60 | 38.11 |
| DeFFusion [6] | | Raw | Semi-supervised | 26.40 | 36.55 |
| Giorgi et al. [3] | Acc. [2], Gyro. [3] | Raw | Fully supervised | 26.36 | 23.28 |
| DeepAuthen [9] | | Raw | Fully supervised | 27.29 | 24.86 |
| Ours | | Raw | Weakly supervised | **18.32** | **22.98** |

[1] Touch screen, [2] accelerometer, [3] gyroscope.

### 5.4. Impact of the Number of impostor Profiles and Sample Size for Training

This experiment analyzes the impact of the number of impostor profiles and samples for training. By default, we use ten samples for each impostor profile, where there are ten profiles in total. Here, we analyze the sensitivity of our model with respect to the number of impostor profiles and samples per profile. The following experiments are based on the HMOG dataset.

Figure 7 illustrates the averaged EERs for the different numbers of impostor profiles involved in training. We observe that the EERs tend to decrease as the number of profiles increases. The LFP-TCN without data augmentation shows EER values of 10.91% and 12.25% when using 40 and 10 impostor profiles, respectively, which represents a performance degradation of 1.34%. On the other hand, the LFP-TCN with the random pattern mixing shows EER values of 9.86% and 10.95% when using 40 and 10 impostor profiles, respectively, in which the performance degradation remains 1.09%. This suggests that the random pattern mixing alleviates the negative impact on performance due to using only a few impostor profiles.

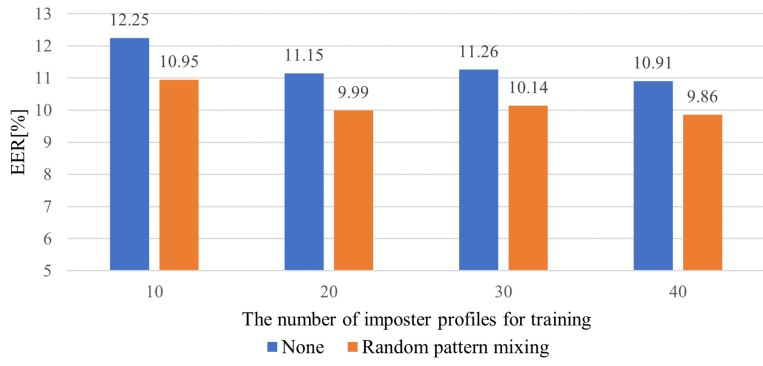

**Figure 7.** Averaged EERs (%) for different numbers of impostor profiles for training.

Figure 8 illustrates the averaged EERs for the different training sample sizes per impostor profile. Similar to Figure 7, the EERs constantly decrease as the sample size increases. The sample size ranges from 10 to 80. The LFP-TCN without data augmentation achieves an EER of 9.64% with 80 samples. Then, the EER degrades to 12.25% with ten samples, which is an increase of 2.61%. On the other hand, the LFP-TCN with random pattern mixing achieves values of EER of 9.33% and 10.95% with 80 and 10 samples, in which the performance degradation remains 1.62%. This shows that the random pattern

mixing improves the sample efficiency, diversifying impostor patterns for the training even with limited impostor data.

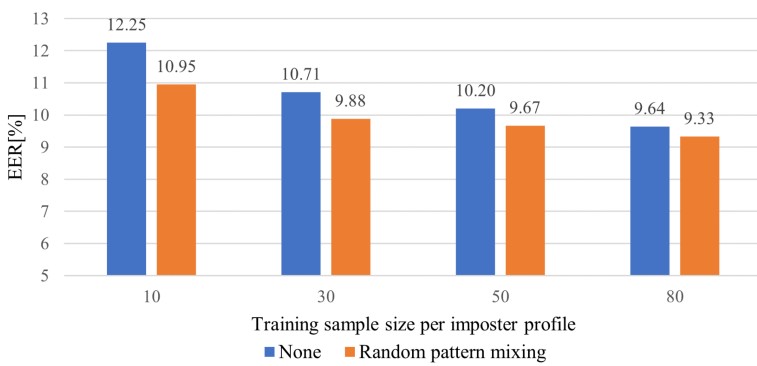

**Figure 8.** Averaged EERs (%) for different training sample sizes per impostor profile.

### 5.5. Impact of Re-Authentication Time

This experiment analyzes the fluctuation of the authentication performance over re-authentication time. We compare the decision, score, and sensor-level fusion. We fix the window size at 1 s for the decision and score-level fusion, while we adjust the window size for the sensor-level fusion according to re-authentication time. We apply this setting to LFP-TCN with random pattern mixing. The following experiments are based on the HMOG dataset.

#### 5.5.1. Decision and Score-Level Fusion

Table 5 shows the averaged EERs of the decision and score-level fusion for different re-authentication time. As shown in Table 5, longer re-authentication time yields significantly better performance. For the re-authentication time of 29 s, the score-level fusion of max-score shows the EER 3.55%. On the other hand, the score-level fusion of the mean score achieves the EER of 5.11%, followed by the decision-level fusion with 5.86%.

**Table 5.** Averaged EERs (%) of the decision and score-level fusion for different re-authentication times.

| Re-Authentication Time | Decision-Level Fusion | Score-Level Fusion | |
| --- | --- | --- | --- |
| | | Mean-Score | Max-Score |
| 1 s | 10.95 | 10.95 | 10.95 |
| 3 s | 9.97 | 8.84 | 8.09 |
| 5 s | 9.15 | 8.00 | 6.98 |
| 7 s | 8.65 | 7.51 | 6.24 |
| 9 s | 8.21 | 7.09 | 5.73 |
| 15 s | 7.21 | 6.26 | 4.81 |
| 19 s | 6.77 | 5.86 | 4.20 |
| 25 s | 6.12 | 5.37 | 3.74 |
| 29 s | 5.86 | 5.11 | 3.55 |

Among these three mechanisms, the score-level fusion of max-score works the best for each re-authentication time. This result indicates that impostors can hardly imitate genuine users completely.

#### 5.5.2. Sensor-Level Fusion

The sensor-level fusion provides one input sample in re-authentication time by enlarging the window size ($T_w = T_r$). To adjust to the longer input sequence, we introduce additional residual blocks and make the receptive field of the LFP-TCN cover the whole sequence.

For the re-authentication time 3 and 5 s, we use 7 residual blocks with the dilation rate of $(1, 2, 4, 8, 16, 32, 64)$ for each block. For the re-authentication time 7 s, we use 8 residual blocks with the dilation rate of $(1, 2, 4, 8, 16, 32, 64, 128)$ for each block.

Table 6 tabulates the number of parameters and the averaged EERs of the sensor-level fusion at different re-authentication times. The EERs decrease for longer re-authentication times; however, the performances are inferior to the decision and score-level fusion. Moreover, the number of model parameters increases according to the re-authentication time, which hinders the use of memory-limited devices. This experiment shows that decision and score-level fusion are preferred for continuous authentication in terms of performance and model complexity.

**Table 6.** Averaged EERs (%) of the sensor-level fusion for different re-authentication time.

| Re-Authentication Time | Sensor-Level Fusion | Parameter Size |
|:---:|:---:|:---:|
| 1 s | 10.95 | 0.5 M |
| 3 s | 10.20 | 0.7 M |
| 5 s | 9.64 | 0.7 M |
| 7 s | 9.68 | 0.8 M |

## 6. Conclusions

This paper proposed a mobile continuous authentication method using the touch-gesture signals acquired from the touch screen, accelerometer, and gyroscope. We posit continuous mobile authentication as a few-shot anomaly detection problem utilizing a novel sequential data augmentation technique and network architecture. We proposed random pattern mixing to diversify impostor features for the training by producing class-unconstrained impostor patterns. Moreover, we presented the local feature pooling-based TCN, a variant of the temporal convolutional network (TCN), aggregating global and local feature information to learn delicate impostor patterns from sensor data. Accordingly, our proposed method can characterize various fine-grained impostor patterns from limited seen impostor samples, gaining high robustness against unseen impostors. Finally, we demonstrated practical continuous authentication based on the decision and score-level fusion. They constantly improve the accuracy performance against extended re-authentication time without causing the long-term dependency problem or increasing model complexity. Our experiments showed the effectiveness of our method compared to baselines and its state-of-the-art performance over the recent seven mobile authentication methods.

Despite the state-of-the-art performance of our method, there is room for further improvement. First, unlike the classical cross entropy loss, a more powerful loss function could be explored, especially for anomaly detection problems. Secondly, how to transfer knowledge from different domains could be investigated to utilize additional prior knowledge of impostors. Lastly, one could design a novel decision-making framework tailored for continuous authentication.

**Author Contributions:** Conceptualization, K.W. and A.B.J.T.; methodology, K.W.; software, K.W.; validation, K.W. and A.B.J.T.; formal analysis, K.W.; investigation, K.W.; resources, K.W.; data curation, K.W.; writing—original draft preparation, K.W.; writing—review and editing, K.W. and A.B.J.T.; visualization, K.W.; supervision, A.B.J.T.; project administration, A.B.J.T.; funding acquisition, A.B.J.T. All authors have read and agreed to the published version of the manuscript.

**Funding:** This work was supported by the National Research Foundation of Korea (NRF) grant funded by the Korea government (MSIP) (NO. NRF-2022R1A2C1010710).

**Institutional Review Board Statement:** Not applicable.

**Informed Consent Statement:** Not applicable.

**Data Availability Statement:** This study leveraged two publicly available datasets, the HMOG [37] and BBMAS [38] datasets. The HMOG dataset is downloadable from https://www.cs.wm.edu/~qyang/

hmog.html (accessed on 20 August 2022). The BBMAS dataset is downloadable from https://ieee-dataport.org/open-access/su-ais-bb-mas-syracuse-university-and-assured-information-security-behavioral-biometrics (accessed on 20 August 2022).

**Conflicts of Interest:** The authors declare no conflict of interest.

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
