# Peer review of "Few-Shot Continuous Authentication for Mobile-Based Biometrics"

_applsci, doi:10.3390/app122010365_

Round 1
Reviewer 1 Report
The topic of this manuscript is very interesting and timely. It tries to solve the problem of the present sub-optimal authentication (one-time) on smartphone with for instance financial transactions. The manuscript is well written.
However, I am not knowledgeable enough in the specific techniques to judge the approach and scientific soundness.
Author Response
Thank you for inviting us to submit a revised draft of our manuscript entitled, “Few-shot Continuous Authentication for Mobile-based Biometrics” to the journal of Applied Sciences. We also appreciate the time and effort you have dedicated to providing insightful feedback on ways to strengthen our paper. Thus, it is with great pleasure that we resubmit our article for further consideration. We have incorporated changes that reflect the detailed suggestions you have graciously provided. We also hope that our edits and the responses we provide below satisfactorily address all the issues and concerns you have noted.
Point 1: “The topic of this manuscript is very interesting and timely. It tries to solve the problem of the present sub-optimal authentication (one-time) on smartphone with for instance financial transactions. The manuscript is well written.”
Response 1: Thank you very much for agreeing with us on the contribution of this manuscript.

Reviewer 2 Report
The authors revised the paper based on the reviewers comment.
Author Response
Thank you for inviting us to submit a revised draft of our manuscript entitled, “Few-shot Continuous Authentication for Mobile-based Biometrics” to the journal of Applied Sciences. We also appreciate the time and effort you have dedicated to providing insightful feedback on ways to strengthen our paper. Thus, it is with great pleasure that we resubmit our article for further consideration. We have incorporated changes that reflect the detailed suggestions you have graciously provided. We also hope that our edits and the responses we provide below satisfactorily address all the issues and concerns you have noted.
Point 1: “The authors revised the paper based on the reviewers comment.”
Response 1: Thank you for your kind reminders. We sincerely revised the manuscript according to the reviewers’ comments.

Reviewer 3 Report
Paper complexity and soundness are adequate. Improvement over early approaches like this ( http://biometrics.cse.msu.edu/Publications/Face/Crouseetal_ContinuousAuthMobileFace_ICB15.pdf) are welcome together with experiments on two public benchmark datasets show the effectiveness of proposed method and its state-of-the-art performance. Accuracy in performance against extended re-authentication time without causing the long-term dependency problem or increasing model complexity is a big and increasing challenge. This research provide robust but more important applicable and efficient method much better then traditional baselines. Comparison with the recent seven mobile authentication methods demonstrate potential and will be great motivation for future research.
Author Response
Thank you for inviting us to submit a revised draft of our manuscript entitled, “Few-shot Continuous Authentication for Mobile-based Biometrics” to the journal of Applied Sciences. We also appreciate the time and effort you have dedicated to providing insightful feedback on ways to strengthen our paper. Thus, it is with great pleasure that we resubmit our article for further consideration. We have incorporated changes that reflect the detailed suggestions you have graciously provided. We also hope that our edits and the responses we provide below satisfactorily address all the issues and concerns you have noted.
Point 1: “Accuracy in performance against extended re-authentication time without causing the long-term dependency problem or increasing model complexity is a big and increasing challenge. This research provide robust but more important applicable and efficient method much better then traditional baselines. Comparison with the recent seven mobile authentication methods demonstrate potential and will be great motivation for future research.”
Response 1: Thank you very much for agreeing with us on the contribution of this manuscript.

Reviewer 4 Report
In this work, the authors propose a novel deep learning-based model, namely Local Feature Pooling based Temporal Convolution Network (LFP-TCN), which directly models raw sequential mobile data, aggregating global and local feature information. The authors, introduce a random pattern mixing augmentation to generate class-unconstrained imposter data for the training. The augmented pool enables characterizing various imposter patterns from limited imposter data. Finally, they demonstrate practical continuous authentication using score-level fusion, which prevents long-term dependency or increased model complexity due to extended re-authentication time.
The authors have addressed a good problem, however, there are few major concerns which need to be addressed before the paper can be accepted for the publication. The major concerns are as follows:
1- Please provide the motivation and contributions of this work as separate sections in the paper for a better understanding of the reader.
2- Please provide some important results in a graphical manner which would help the reader to get more information at a single glance. At present all the results are illustrated in tabular form.
3- Please provide the possible future works which can be considered as an extension of this work.
The authors are encouraged to revise and resubmit the work.
Author Response
Thank you for inviting us to submit a revised draft of our manuscript entitled, “Few-shot Continuous Authentication for Mobile-based Biometrics” to the journal of Applied Sciences. We also appreciate the time and effort you have dedicated to providing insightful feedback on ways to strengthen our paper. Thus, it is with great pleasure that we resubmit our article for further consideration. We have incorporated changes that reflect the detailed suggestions you have graciously provided. We also hope that our edits and the responses we provide below satisfactorily address all the issues and concerns you have noted.
Point 1: “Please provide the motivation and contributions of this work as separate sections in the paper for a better understanding of the reader.”
Response 1: Thank you for your kind reminders. We added a subsection "Motivation and contribution" (section 1.1) under the section “Introduction”.
Point 2: “Please provide some important results in a graphical manner which would help the reader to get more information at a single glance. At present all the results are illustrated in tabular form.”
Response 2: Thank you for your kind reminders. In the section “Experimental result”, we replaced four tables with bar graphs for better visualization, i.e., Figures 5, 6, 7, and 8.
Point 3: “Please provide the possible future works which can be considered as an extension of this work.”
Response 3: Thank you for your kind reminders. We added three suggestions for future directions in the section “Conclusion”.
Again, thank you for giving us the opportunity to strengthen our manuscript with your valuable comments and queries. We have worked hard to incorporate the reviewers’ feedback and hope that these revisions persuade you to accept our submission.

Round 2
Reviewer 4 Report
The authors have addressed all my concerns. The paper can be accepted for publications in its current form.